# The gut microbial metabolic capacity of microbiome-humanized *vs*. wild type rodents reveals a likely dual role of intestinal bacteria in hepato-intestinal schistosomiasis

**Alba Cortés[1,2], John Martin[3], Bruce A. Rosa[3], Klara A. Stark[1], Simon Clare[4,5], Catherine McCarthy[5], Katherine Harcourt[5], Cordelia Brandt[5], Charlotte Tolley[5,6], Trevor D. Lawley[5], Makedonka Mitreva[3], Matthew Berriman[5¤a]\*, Gabriel Rinaldi[5☯¤b]\*, Cinzia Cantacessi[1☯]\***

**1** Department of Veterinary Medicine, University of Cambridge, Cambridge, United Kingdom, **2** Departament de Farmàcia, Tecnologia Farmacèutica i Parasitologia, Facultat de Farmàcia, Universitat de València, Burjassot, València, Spain, **3** Division of Infectious Diseases, Department of Medicine, Washington University School of Medicine, St. Louis, Missouri, United States of America, **4** Department of Medicine, University of Cambridge, Cambridge, United Kingdom, **5** Wellcome Trust Sanger Institute, Wellcome Genome Campus, Hinxton, United Kingdom, **6** Cambridge Institute of Therapeutic Immunology and Infectious Disease, University of Cambridge, Cambridge, United Kingdom

☯ These authors contributed equally to this work.
¤a Current address: Institute of Infection, Immunity and Inflammation, University of Glasgow, Glasgow, United Kingdom
¤b Current address: Department of Life Sciences, Aberystwyth University, Aberystwyth, United Kingdom
\* matt.berriman@glasgow.ac.uk (MB); gabriel.rinaldi@aber.ac.uk (GR); cc779@cam.ac.uk (CC)

**Data Availability Statement:** The sequencing data generated in this study are available in the

## Abstract

Increasing evidence shows that the host gut microbiota might be involved in the immunological cascade that culminates with the formation of tissue granulomas underlying the pathophysiology of hepato-intestinal schistosomiasis. In this study, we investigated the impact of *Schistosoma mansoni* infection on the gut microbial composition and functional potential of both wild type and microbiome-humanized mice. In spite of substantial differences in microbiome composition at baseline, selected pathways were consistently affected by parasite infection. The gut microbiomes of infected mice of both lines displayed, amongst other features, enhanced capacity for tryptophan and butyrate production, which might be linked to the activation of mechanisms aimed to prevent excessive injuries caused by migrating parasite eggs. Complementing data from previous studies, our findings suggest that the host gut microbiome might play a dual role in the pathophysiology of schistosomiasis, where intestinal bacteria may contribute to egg-associated pathology while, in turn, protect the host from uncontrolled tissue damage.

## Author summary

Schistosomiasis is a neglected tropical disease affecting >250 million people worldwide. Causative agents are parasitic worms of the genus *Schistosoma*, that inhabit the small

European Nucleotide Archive (ENA) with the study accession number PRJEB40471.

**Funding:** This research was funded by grants awarded by the Wellcome Trust [grant 206194] to MB and GR, the Fundación Alfonso Martín Escudero to AC and the University of Cambridge to CC. KAS is the grateful recipient of a PhD scholarship by the Cambridge Trust. The funders had no role in study design, data collection and analysis, decision to publish, or preparation of the manuscript.

**Competing interests:** The authors declare that they have no competing interests

blood vessels irrigating the gut and urinary bladder. Adult schistosomes lay their eggs that make their way to the environment by piercing the intestinal or urinary bladder wall to reach the host feces and urine, respectively. *Schistosoma* infections are associated with substantial alterations of gut bacterial communities of both naturally infected humans and experimentally infected mice. Intestinal bacteria are also hypothesized to contribute to the intestinal pathology caused by migrating parasite eggs. Here, we investigated the impact of schistosome infection on the composition and function of gut bacteria inhabiting wild type and microbiome-humanized laboratory rodents. Schistosome colonization was consistently associated with an increased (predicted) ability of gut bacteria to synthetize tryptophan and butyrate, both of which are known to promote intestinal barrier function by preventing translocation of potentially harmful bacteria into the circulation. Our findings suggest that intestinal bacteria may exert both harmful and protective roles during schistosomiasis, and provide a basis for the development of much needed novel and sustainable strategies for management and control of this neglected disease.

## Introduction

Schistosomiasis, caused by blood flukes of the genus *Schistosoma*, is a neglected tropical disease affecting the world's poorest communities [1]. Globally, >250 million people are infected with at least one of the three major species that infect humans—*S. mansoni*, *S. japonicum* and *S. haematobium*—mostly in regions of sub-Saharan Africa and focal areas of America, the Middle East and Asia [1]. Adult male and female worms dwell *in copula*, first in the portal system, and then, in the mesenteric or pelvic veins of their hosts. Fertilized females lay eggs that, following a migration through the lining of small venules, pierce the wall of the intestine (*S. mansoni* and *S. japonicum*) or the bladder (*S. haematobium*) to be released with the host feces or urine, respectively [1]. Subsequently, when eggs reach the freshwater environment, they hatch ciliated miracidia that seek out and infect snail intermediate hosts [1]. Inside the snail, the parasites undergo a phase of asexual reproduction, i.e., from mother and daughter sporocysts to free-living infectious cercariae that leave the snail to swim and seek the definitive mammalian host. The cercariae penetrate the host skin, shed their tails and become schistosomula that enter the circulation and migrate through the lungs, heart and liver, where they further develop into male and female worm pairs. Between 3 and 4 weeks post-infection, paired adult worms migrate to the mesenteric venules (*S. mansoni* and *S. japonicum*) or the venous plexus of the urinary bladder (*S. haematobium*) and begin to release hundreds of eggs per day [1]. The pathophysiology of acute schistosomiasis is mainly associated with parasite eggs becoming trapped in various organs and tissues (i.e., the intestine and liver, or the urinary bladder) and subsequent formation of inflammatory granulomas containing immune cells (alternatively activated macrophages, granulocytes, Th2 and B cells) and extracellular matrix [2]. During the chronic phase of infection, granulomas shrink and are replaced by fibrous tissue, that often compromises the function of affected organs [1].

The genesis of *Schistosoma* egg-induced granulomas is the result of a finely regulated crosstalk between egg-secreted antigens and host immunity [2,3]. However, over the last few years, evidence has emerged of the likely contribution of a third player–the host gut microbiota–in the cascade of immunological events that culminate with the formation of tissue granulomas [4–8]. First, in a seminal study conducted in murine schistosomiasis [4], administration of broad-spectrum antibiotics and antimycotics resulted in substantially decreased intestinal inflammation and granuloma development. In the same study, lymphocyte preparations from

mesenteric lymph nodes retrieved from microbiota-depleted mice showed reduced production of IFN-γ and IL-10 when exposed to *S. mansoni* secreted egg antigen, thus indicating that gut resident bacteria may influence schistosome-specific immunity [4]. Supporting the hypothesis of a key role of the host gut microbiota in the pathophysiology of schistosomiasis, we previously detected dramatic alterations in the gut microbial profiles of mice during acute patent infection by *S. mansoni* compared with uninfected controls [5]; these alterations included substantially decreased and increased gut bacterial alpha- and beta diversity, respectively, as well as changes in the relative abundances of populations of microbes with putative roles in host immune-regulatory and/or inflammatory responses [5]. We therefore speculated that such changes, indicative of dysbiosis and accompanying egg migration across the gut wall and consequent immunopathology, might contribute to the pathophysiology of hepato-intestinal schistosomiasis [5]. In a subsequent study, we attempted to investigate the translational significance of these findings by characterizing the gut microbial profiles of microbiota-humanized [human-microbiota-associated (HMA)] and wild type (WT) mice prior to and following infection with *S. mansoni* [8]. High-throughput sequencing of the bacterial 16S rRNA gene revealed profound differences between the microbial profiles of these cohorts at baseline, with the gut microbiota of HMA being characterized by low microbial alpha diversity, expanded populations of Proteobacteria and absence of lactobacilli compared to WT rodents [8]. Of note, whilst *S. mansoni* infection was associated with a significant decrease in gut microbial alpha diversity in both rodent lines, analysis of bacterial taxa whose relative abundances were altered following worm colonization revealed substantial dissimilarities between HMA and WT mice [8]. Interestingly, significantly higher worm and egg burdens were recovered from HMA *vs.* WT at the end of the experiment [8]. Based on this observation, we hypothesized that the structure and function of the host baseline gut microbiome (i.e., prior to infection) might play a key role in the complex network of interactions between schistosome parasites, the host and its immune system. Whilst data from these studies represent a step forward in our current understanding of these relationships, the microbiota profiling techniques applied thus far (i.e., bacterial culturing followed by colony counting and bacterial 16S rRNA gene amplicon sequencing) [4,5,8] do not permit explorations of schistosomiasis-associated changes in gut bacterial populations beyond genus-level taxonomic mapping. Determining differences in gut microbial communities between schistosome-infected and -uninfected HMA and WT mice at species level, and evaluating the relative contribution of each community member to the overall pool of functional genes, are nonetheless pivotal. This knowledge is likely to unveil mechanisms underlying crosstalk between host, parasite and microbiome during schistosome infection.

In this study, we undertake high-throughput shotgun metagenomics sequencing of the gut microbiome of HMA and WT mice to provide a high-resolution map of the functional potential of gut bacterial communities established in these mouse lines. Furthermore, we investigate changes in the composition and functional potential of these communities following experimental infection with *S. mansoni*. We show that, despite profound differences in compositional and functional profiles between the two murine lines, parasite colonization induced partially overlapping alterations in bacterial metabolic capacity, including up-regulation of pathways linked to L-tryptophan biosynthesis and fermentation to short chain fatty acids (SCFAs).

## Results and discussion

### Gut bacterial profiles differ between WT and HMA mice

Whole genome sequencing (WGS) of fecal DNA extracts from both *S. mansoni*-infected (hereafter referred to as '*Sm+*') and uninfected ('*Sm-*') WT and HMA mice yielded $228.03 \cdot 10^6$ ±

$31.13 \cdot 10^6$ (mean ± standard deviation) reads per sample, while no-DNA template negative controls generated $10.56 \cdot 10^3 ± 13.27 \cdot 10^3$ reads (S1 Fig). Mean base count per sample prior to and following quality filtering and removal of host sequences did not yield significant differences between mouse lines, nor between infection groups within each line (S1 Fig).

Taxonomic profiling of filtered metagenomic data identified a total of 449 bacterial species across samples encompassing both WT (418 species) and HMA mice (363 species); these belonged to 187 genera, of which 38 and 18 were unique to WT and HMA, respectively (S1 Table and S2 Fig). Substantial differences in both bacterial species composition and corresponding relative abundances were observed between the gut microbiomes of WT and HMA mice (Figs 1, S2 and S3), with HMA gut bacterial communities clustering separately from those of WT by Principal Coordinates Analysis (PCoA) (Fig 2A), as previously reported [8]. Moreover, microbial beta diversity in WT was significantly higher than that in HMA (cf. Fig 2A; ANOSIM: R = 1, p<0.001), likely due to differences in breeding and housing conditions between these rodent lines (i.e., specific-pathogen-free environment for WT *vs.* isolation for HMA).

Species richness and alpha diversity (Shannon index) in the gut microbiome of WT were significantly higher than those in HMA (t-test p = 0.002, $t$ = 3.362, df = 26; and p<0.0001, $t$ = 6.450, df = 26, respectively) (Fig 2B). This observation is in accordance with our previous marker-gene-based work, where significantly higher bacterial alpha diversity was observed in the gut microbiota of WT compared to HMA mice [8], a finding tentatively attributed to a likely failure of selected human gut microbes to colonize the mouse gastrointestinal (GI) tract [8]. Moreover, according to previous observations [8], substantial differences were detected in the relative abundances of predominant taxa in the GI tracts of WT and HMA (S2 Fig). For instance, several *Bacteroides* species, including *B. cellulosilyticus*, *B. vulgatus*, *B. dorei* and *B. thetaiotaomicron*, amongst others, predominated in HMA, whereas *B. acidifaciens* was significantly more abundant in WT mice. These differences, along with the absence of individual bacterial species (and/or genera) in the gut of WT or HMA mice, might impact the overall functional potential of the microbiomes of these mice, as well as of the microbiota-mediated immune modulation [9]. A striking example is represented by the genus *Lactobacillus*, whose members were only detected in the gut of WT animals (S1 Table). These bacteria are common inhabitants of mouse and human GI tracts, and their relative abundance frequently increases following microbial transfers between mice [10,11]; however, previous studies have reported a failure of members of *Lactobacillaceae* to engraft the colons of HMA mice following fecal material transplant of human feces containing lactobacilli [12]. Most lactobacilli produce lactate, that may be further fermented by other gut bacteria into butyrate, a SCFA with anti-inflammatory properties [13,14]. In addition, selected species and strains of *Lactobacillus* are responsible for activation of mechanisms of innate immunity in the host GI tract *via* toll-like receptor (TLR)-dependent and -independent pathways [15,16]. Interestingly, our previous data revealed higher *S. mansoni* infection burdens in HMA *vs.* WT mice, which led us to hypothesize the occurrence of a link between host baseline microbiota composition and susceptibility to infection [8]. Whilst attributing these observations to the absence of lactobacilli in the HMA gut is currently unwarranted, our data point toward this bacterial group as a potential target for future mechanistic studies aimed to elucidate these interactions.

## The gut bacterial metagenomes of WT and HMA mice feature distinct functional profiles

We then performed functional annotations of gut metagenome data of both *Sm*+ and *Sm*- WT and HMA mice using established bioinformatics pipelines (see Materials and Methods).

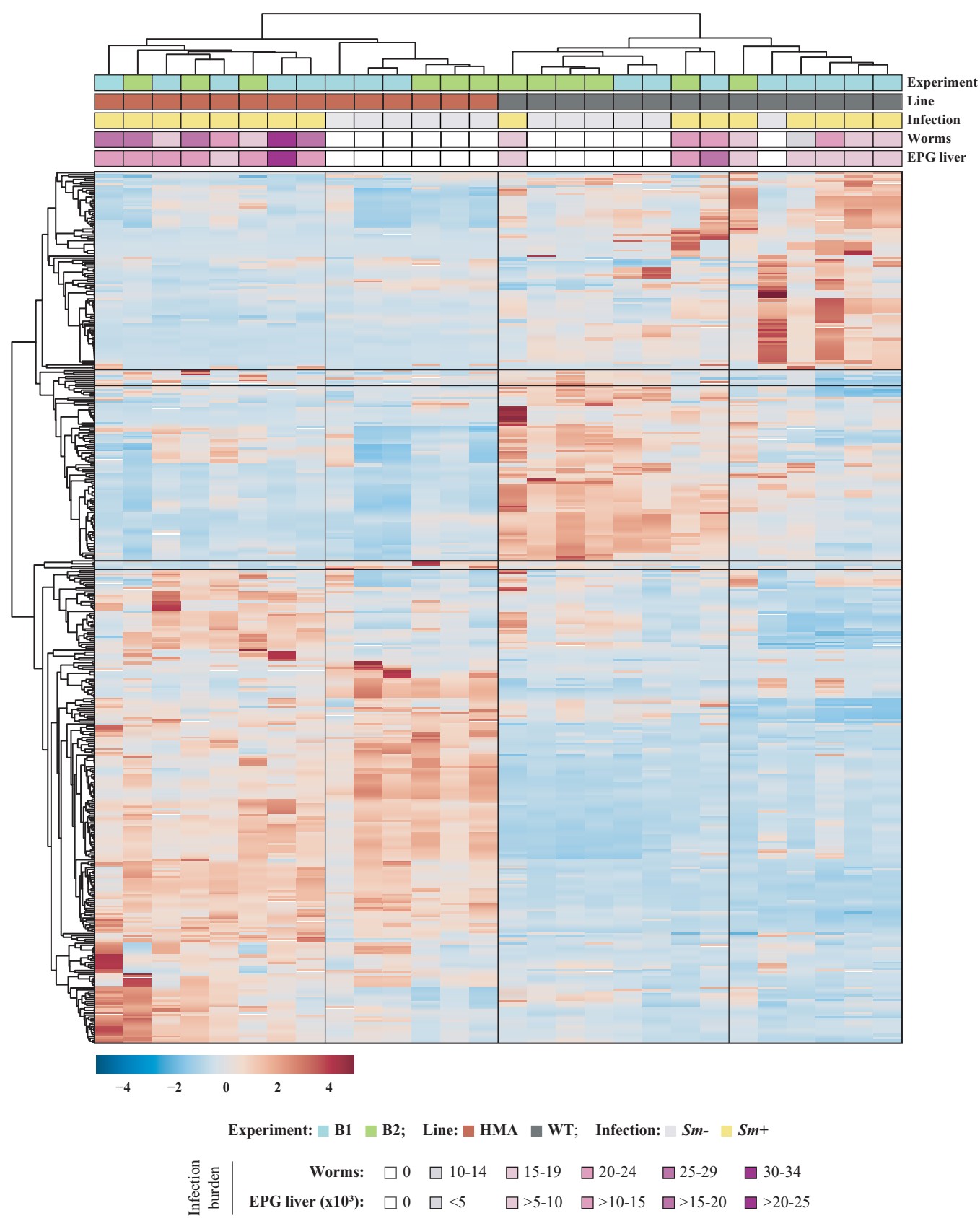

**Fig 1. Gut bacterial community composition.** Hierarchical clustering of gut bacterial communities of wild type (WT) and human-microbiota-associated (HMA) mice (columns) grouped according to Pearson correlation between species relative abundances (rows) and explanatory variables indicated at the top of the figure [i.e., experiment, mouse line, infection status and infection burden (EPG = eggs per gram)]. Dark lines on the heatmap separate major clusters in both axes.

Principal Component Analysis (PCA) applied to pathway abundance data separated the gut microbiomes of WT and HMA mice into two distinct clusters along PC1 (Fig 2C) [supported by Canonical Correspondence Analysis (CCA; p = 0.001, F = 64.9)], thus revealing profound differences in the overall functional profiles of bacterial communities colonizing the gut of each murine line.

A total of 289 metabolic pathways were identified by HUMAnN3 (https://huttenhower.sph.harvard.edu/humann/) in at least five samples of WT and/or HMA mice; of these, 244 and 276 were detected in the metagenomes of WT and HMA, respectively, and analyzed further (S4 Fig and S2 Table). Thirteen pathways were solely detected in the microbiomes of WT; these included, amongst others, 'Bifidobacterium shunt' (P124-PWY; a fermentation pathway for production of lactate and the SCFA acetate), 'L-glutamate degradation IV' (PWY-4321) and 'spermine and spermidine degradation I' (PWY-6117) (S2 Table). Notably, bacterial species contributing to pathways solely identified in WT remained mostly unclassified, except for *Lactobacillus reuteri*, which was associated with 'L-glutamate degradation IV' (PWY-4321), 'superpathway of geranylgeranyldiphosphate biosynthesis I (via mevalonate)' (PWY-5910) and 'mevalonate pathway I (eukaryotes and bacteria)' (PWY-922) (S2 Table). HUMAnN relies on the MetaPhlAn tool (https://huttenhower.sph.harvard.edu/metaphlan/) to identify bacterial organisms present in a given metagenome [17]. The MetaPhlAn algorithm estimates microbial taxa abundance based on the coverage of clade-specific marker genes that are present in all strains of a given species, thus representing a powerful strategy to generate unambiguous markers for genetic characterization of metagenomic species [17]. Such an approach, however, favors the identification of species with several strains for which complete genomes are available; thus, taxonomic annotation using this approach is likely to lead to the identification of a smaller number of species compared with WGS read-mapping against reference genomes in the GenBank database, as the latter contains sequences from several species with incomplete genome assemblies.

The 45 pathways uniquely identified in the microbiomes of HMA included eight superpathways for the biosynthesis of menaquinones [e.g., 'superpathway of menaquinol-8 biosynthesis I' (PWY-5838) and 'superpathway of menaquinol-11 biosynthesis' (PWY-5897)], several pathways involved in the synthesis of cell wall and outer membrane components ['peptidoglycan biosynthesis IV (*Enterococcus faecium*)' (PWY-6471), 'peptidoglycan biosynthesis V (β-lactam resistance)' (PWY-6470), 'peptidoglycan biosynthesis II (staphylococci)' (PWY-5265), 'superpathway of UDP-glucose-derived O-antigen building blocks biosynthesis' (PWY-7328) and 'superpathway of dTDP-glucose-derived O-antigen building blocks biosynthesis' (PWY-7317)], and pathways for the degradation/utilization of carbohydrates, including 'trehalose degradation V' (PWY-2723), 'glycogen degradation I' (GLYCOCAT-PWY), 'starch degradation III' (PWY-6731) and 'glucose and glucose-1-phosphate degradation' (GLUCOSE1PME-TAB-PWY). Similar to WT mice, bacterial species contributing to HMA-unique pathways remained largely unclassified (S2 Table).

Differences between the gut bacterial profiles of WT *vs*. HMA are likely to underpin profound functional dissimilarities between these communities. Other external factors that had shaped the composition of the donor microbiome inoculum prior to administration to HMA recipients (e.g., diet, feeding patterns and metabolic requirements, among others) might also contribute to these functional differences [18]. However, it is important to point out that such

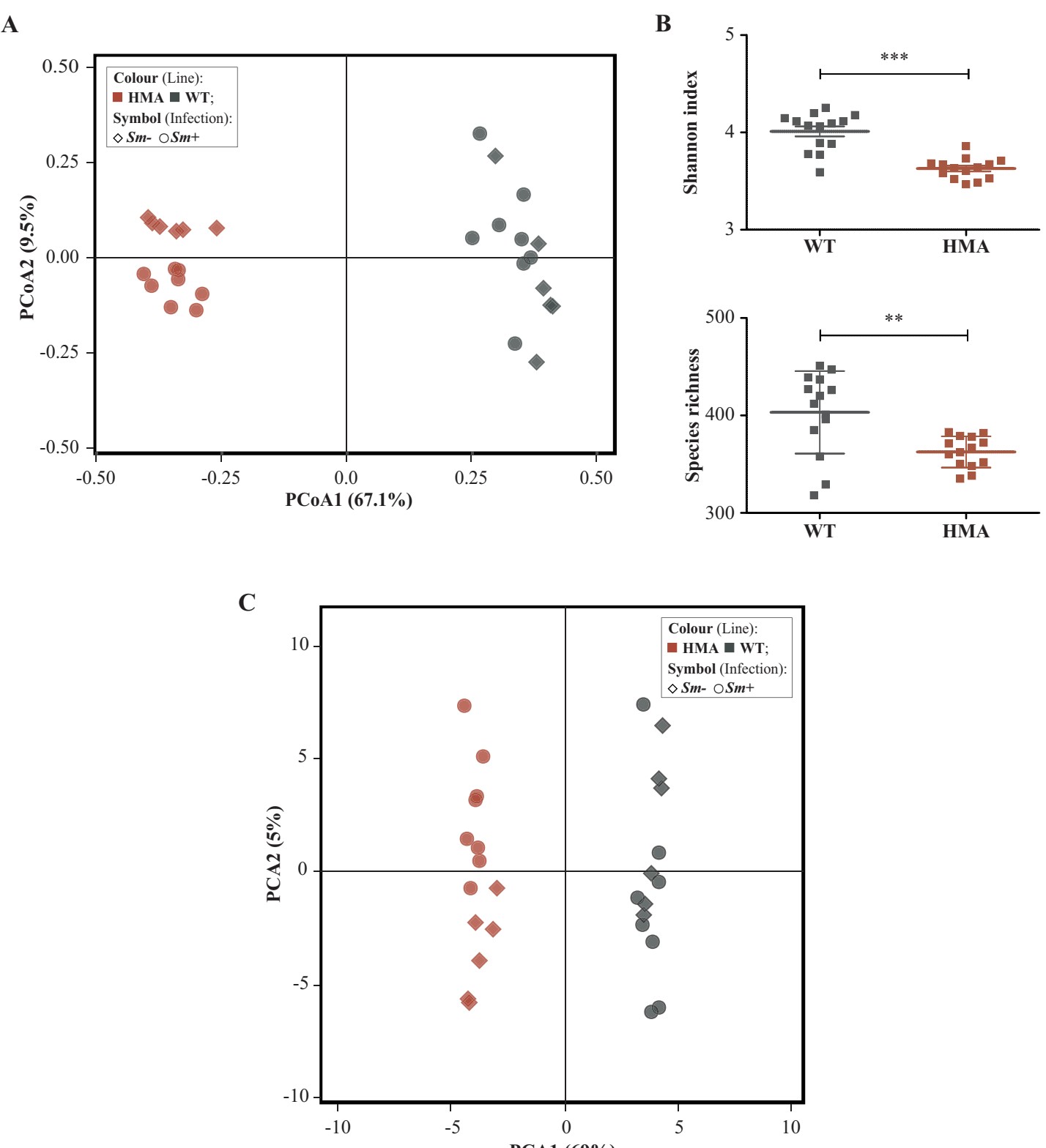

**Fig 2. Compositional and functional profiles of the gut metagenomes of wild type (WT) and human-microbiota-associated (HMA) mice.** (A) Principal Coordinates Analysis (PCoA) of microbial community composition of all *Schistosoma*-infected and -uninfected animals in both experiments, determined by mapping of shotgun metagenomic reads against bacterial reference genomes available from the GenBank database. (B) Alpha diversity. (C) Principal Component Analysis (PCA) of pathway relative abundances at community-level as determined by HUMAnN3. Asterisks indicate significant differences between groups determined by unpaired t-test: **p<0.01; ***p<0.0001.

discrepancies may not be conclusively associated to the differences in *S. mansoni* infection burdens observed in WT *vs.* HMA mice [cf. 8], as similar end-products of microbiome metabolism could be generated *via* distinct biological pathways. For instance, the 'Bifidobacterium shunt' pathway detected in WT mice might be associated with enhanced ability to produce lactate/acetate (and thus, butyrate) [13,14]; however, several HMA-unique pathways for carbohydrates degradation (see above) might result in higher rates of pyruvate biosynthesis and, in turn, of SCFA production [19].

### *Schistosoma mansoni* infection is associated with altered gut bacterial community structure and species composition in both WT and HMA mice

We subsequently focused on characterizing the effect(s) of *S. mansoni* infection on the gut bacterial community structure of each WT and HMA. Using PCoA, gut bacterial communities of both WT and HMA clustered by infection status, although only HMA *Sm+* and *Sm-* samples grouped separately along the PCoA1 axis (Fig 3A and 3B). Microbial beta diversity was significantly higher across *Sm+* samples of both WT (ANOSIM: R = 0.361, p<0.004) and HMA (ANOSIM: R = 0.651, p<0.002) compared to their uninfected counterparts. In contrast, Shannon index was unaltered following infection of either line, although a significant decrease in gut bacterial richness was observed in HMA at 50 days post *S. mansoni* colonization (t-test p = 0.014, $t$ = 2.864, df = 12) (S5 Fig). Infections by *Schistosoma* spp. had been associated with significantly decreased gut bacterial richness and alpha diversity, and linked to local inflammation and tissue damage caused by migrating parasite eggs [5,7]. Differences in infection burdens [cf. 5] or *Schistosoma* species [cf. 7] used in these previous studies might hold partially accountable for this discrepancy. Nonetheless, our published 16S rRNA gene amplicon sequencing data had also revealed significant reductions in alpha diversity in WT and HMA mice infected with *S. mansoni* [8]. Hence, our current observations might be indicative of the fact that marker gene data may lead to an overestimation of the impact of the infection on gut bacterial alpha diversity.

Hierarchical clustering of WT and HMA gut microbiomes grouped HMA samples according to infection status, while WT *Sm+* and *Sm-* samples remained partially intermingled; moreover, ordination of infected samples according to infection burdens [i.e., worm counts and/or eggs per gram (EPG) of liver] was not observed in either mouse line (Fig 1). In order to gain insights into species whose abundances were altered following *S. mansoni* infection, the relative abundances of identified bacteria were compared between *Sm+* and *Sm-* mice of each line. Significant differences in bacterial species abundance associated with infection status were detected in both WT and HMA mice by Linear Discriminant Analysis Effect Size (LEfSe) and confirmed by Mann-Whitney *U* test; these alterations encompassed members of the genera *Bacteroides*, *Parabacteroides*, *Alistipes* and *Barnesiella* (Fig 3C and S3 Table), which had been previously reported to change in response to *S. mansoni* infection in WT and/or HMA mice in targeted metagenomics studies [5,6,8]. Furthermore, strong significant correlations (i.e., Pearson's $r \geq |0.8|$, p<0.001) were observed between selected bacterial species significantly impacted by infection (cf. Fig 3C) and parasite burdens (i.e., worm counts and/or EPG of liver; cf. [8]) in the two mouse lines (S4 Table).

Both *Bacteroides* and *Alistipes* harbor species with known anti-inflammatory properties, whilst expansions of selected members of each genus have been also linked to selected inflammatory conditions. For instance, *B. acidifaciens*, a species expanded in the gut of WT *Sm+* mice, has been reported to overgrow following IL-22 blockage in a mouse model of chronic *Salmonella* gastroenteritis, ultimately leading to exacerbated intestinal inflammation [20]. Additional studies, however, have demonstrated protective effects of this bacterial species

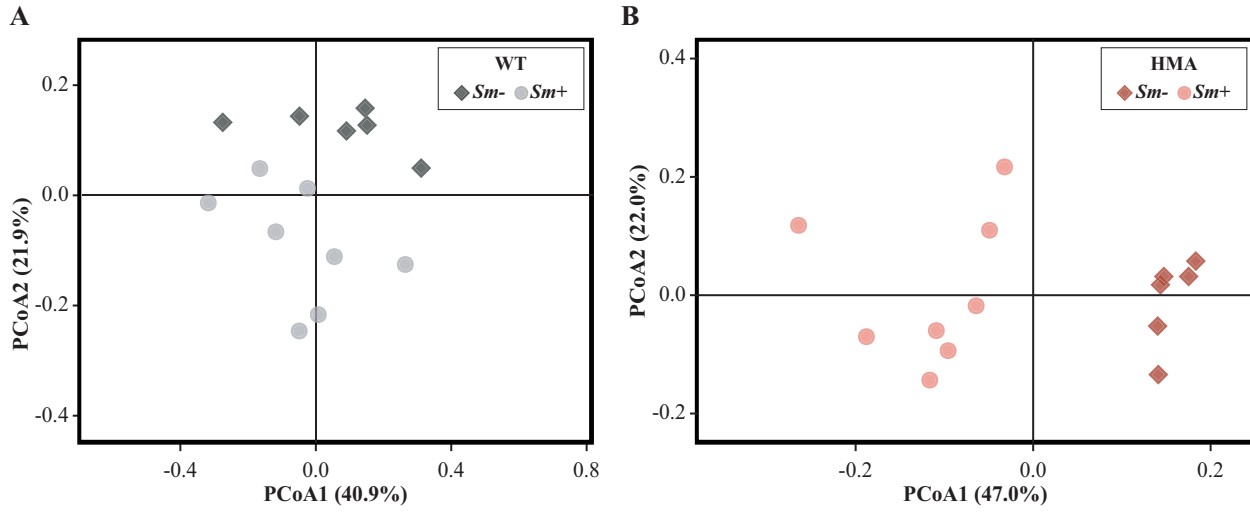

| WT Bacterial species | Associated group | |
|---|---|---|
| | Sm+ | Sm- |
| *Bacteroides acidifaciens* | ▨ | |
| *Bacteroides caecimuris* | ▨ | |
| *Bacteroides caecimuri* | ▨ | |
| *Alistipes dispar* | ▨ | |
| *Alistipes shahii* WAL 8301 | ▨ | |
| *Alistipes senegalensis* JC50 | ▨ | |
| *Alistipes finegoldii* DSM 17242 | ▨ | |
| *Parabacteroides distasonis* ATCC 8503 | ▨ | |
| *Alistipes provencensis* | ▨ | |
| *Lactobacillus reuteri* DSM 20016 | ▨ | |
| *Lactobacillus johnsonii* NCC 533 | ▨ | |
| *Alistipes megaguti* | ▨ | |
| *Alistipes timonensis* JC136 | ▨ | |
| bacterium 1xD42 87 | | ▨ |
| bacterium D16 29 | | ▨ |
| *Muribaculaceae bacterium* | | ▨ |
| *Schaedlerella arabinosiphila* | | ▨ |

**HMA**

| Bacterial species | Associated group | |
|---|---|---|
| | Sm+ | Sm- |
| *Parbacteroides goldsteinii* DSM 19448 = WAL 12034 | ▨ | |
| *Barnesiella intestinihominis* YIT 11860 | ▨ | |
| *Bacteroides sartorii* | ▨ | |
| *Bacteroides uniformis* | | ▨ |
| *Bacteroides cellulosilyticus* | | ▨ |
| *Bacteroides rodentium* JCM 16496 | | ▨ |
| *Bacteroides timonensis* | | ▨ |
| *Bacteroides xylanisolvens* | | ▨ |
| *Bacteroides intestinalis* | | ▨ |
| *Bacteroides stercorirosoris* JCM 17103 | | ▨ |

LDA score (log10)   <2.5   2.5-3.0   >3

**Fig 3. Alterations in gut microbial community composition of wild type (WT) and human-microbiota-associated (HMA) mice associated with *Schistosoma mansoni* infection.** Principal Coordinates Analyses (PCoA) of microbial community profiles of WT (A) and HMA (B) mice infected with *S. mansoni* (Sm+) and uninfected controls (Sm-). (C) Bacterial species displaying significant differences in abundance between infected and uninfected samples of each WT and HMA mice, based on Linear Discriminant Analysis Effect Size (LEfSe).

against liver inflammation and allergic asthma that may be mediated by the production of SCFAs [21,22]. Remarkably, expansion of *B. acidifaciens* and increased butyrate production were linked to enhanced gut barrier integrity and function. Similarly, while several *Alistipes* species have been linked to the gut of healthy and diseased individuals [23], oral gavage with *A. finegoldii* (significantly expanded in WT Sm+) was associated with protection against colitis in mice [24].

Some *Bacteroides* species, whose abundances were reduced upon *S. mansoni* infection in HMA, have been reported to exert anti-inflammatory effects in rodent models of intestinal inflammation, and/or when administered as probiotics; e.g., *B. uniformis* [25,26], and *B. cellulosilyticus* [27]. However, intriguingly, these changes were accompanied by a significant expansion of *P. goldsteinii* (Fig 3C), whose relative abundance had been negatively correlated with fecal levels of lipocalin-2 (a marker of intestinal inflammation) in mice genetically susceptible to colitis [28]. This association suggested a protective effect of *P. goldsteinii* against colitis, which was tentatively attributed to the ability of this bacteria to produce SCFAs [29], as well as to promote regulatory IL-10 and reduce proinflammatory IL-1β and intestinal permeability [30,31]. Expansions of the genus *Parabacteroides* were previously linked to *S. mansoni* infection in both HMA and WT mice [8]. However, our current data show that, unlike in HMA, expansion of this genus in the gut of WT Sm+ mice was mainly linked to *P. distasonis* (Fig 3C), whose roles in health and disease may vary depending on bacterial strain- and host-related factors [32].

Additionally, previously unreported expansions of the *Lactobacillus* species *L. reuteri* and *L. johnsonii* were observed in the gut microbiomes of WT Sm+ mice compared with uninfected counterparts (Fig 3C). Alterations of GI populations of lactobacilli have been described in several rodent models of helminth colonization, including mice infected with *S. mansoni* [reviewed in 33]. In particular, a transient expansion of *Lactobacillus* spp. was reported in mice prior to the onset of egg laying by *S. mansoni*, that was however not maintained post-egg laying [5]. Likewise, our previously published targeted metagenomics data did not reveal significant alterations in the abundance of *Lactobacillus* post egg-laying [8]. Hence, current data pointing towards expanded populations of selected lactobacilli following *S. mansoni* infection might be indicative of a role of these bacteria in the pathophysiology of schistosomiasis, and are thus worth exploring. Indeed, the roles of lactobacilli in mediating protection against GI inflammation have been extensively described [34,35]. For instance, *L. reuteri* exerts protective effects against chemically-induced experimental colitis *via* reduced leukocyte recruitment, platelet-mediated inflammation and bacterial translocation [36,37]; similarly, *L. johnsonii* has been demonstrated to exert protective anti-inflammatory activity against bacterial colitis in mice [38]. Moreover, the occurrence of a mutualistic relationship between gut lactobacilli and the GI nematode *Heligmosomoides polygyrus* was proposed in a seminal study showing that supplementation of *L. taiwanensis* promoted chronic helminth establishment *via* Treg expansion in rodent mesenteric lymph nodes [39]. Whether *L. reuteri* and/or *L. johnsonii* play a role in regulating host immunity and/or local inflammation in hepato-intestinal schistosomiasis is yet to be determined. However, remarkably, loss of lactobacilli in antibiotic-treated mice recolonized by co-housing with *S. mansoni*-infected rodents resulted in increased susceptibility to DSS-induced colitis, when compared to microbiota-depleted mice co-housed with uninfected controls [6].

Altogether, observed changes in the composition of the gut microbiota at species-level resolution suggest that several bacteria might be selectively expanded in the gut of WT and HMA mice following *S. mansoni* infection, and assist with protection against egg-mediated tissue damage and inflammation. However, given that the impact of *S. mansoni*-induced intestinal pathology and immune environment on the protective effects of some of these bacteria is difficult to assess [20], and that such protective effects might be associated with specific strains of these bacterial species [32], this hypothesis remains to be thoroughly tested. Indeed, not only determining individual species whose abundances are affected by *S. mansoni* infection, but also their contribution to intestinal protection/damage is essential to gain new insights onto the role(s) of gut bacteria in the pathophysiology of schistosomiasis.

## Schistosomiasis is associated with alterations of the gut microbial functional profiles of WT and HMA mice

The gut bacterial metagenomes of HMA and WT mice were clustered according to their predicted metabolic profiles by pathway abundance data, stratified by known and unclassified organisms. PCA revealed marked differences between the gut bacterial functional profiles of HMA *Sm-* and HMA *Sm+*, that were supported by CCA (F = 5.29, p = 0.001; Fig 4A). In contrast, WT samples clustered by experiment (S6 Fig), rather than by infection status (Fig 4A), with minimal overlap. Accordingly, CCA yielded significant differences in WT gut microbial metabolic profiles according to experiment (F = 3.05, p = 0.002), but not to infection (F = 1.49, p = 0.128).

Metabolic pathways detected in at least 5 metagenomes of either HMA or WT were analyzed for correlation with *S. mansoni* infection by MaAsLin2 and LEfSe on unstratified data. The complete lists of metabolic pathways associated with the gut metagenomes of *Sm+* and *Sm-* of each WT and HMA, by MaAsLin2 and/or LEfSe, are available from S5 and S6 Tables, respectively. A total of 31 metabolic pathways were consistently associated with *S. mansoni* infection in WT mice by both MaAsLin2 (p<0.05; FDR q-value>0.05 for all features) and LEfSe [Linear Discriminant Analysis (LDA) score (log)>2]. In particular, 21 pathways were overrepresented in the gut microbiomes of WT *Sm+*, and 10 in the microbiomes of WT *Sm-* (S5 Table). In HMA, 50 metabolic pathways were consistently associated with *S. mansoni* infection by both MaAsLin2 and LEfSe (S6 Table). Of these, 19 were overrepresented in the gut microbiomes of HMA *Sm+* mice, and 31 in the microbiomes of HMA *Sm-*. In both mouse lines, most of the metabolic pathways altered by *S. mansoni* infection could be grouped into three functional categories: (i) biosynthesis, (ii) degradation/utilization/assimilation and/or (iii) generation of precursor metabolites and energy (S5 and S6 Tables). Eleven metabolic pathways displayed significant changes with *S. mansoni* infection in both WT and HMA (S7 Table); of these, 3 were consistently altered by infection in both WT and HMA, including a pathway for the biosynthesis of L-tryptophan (i.e., TRPSYN-PWY) (Fig 4B). Moreover, the gut microbiomes of both WT *Sm+* and HMA *Sm+* displayed an enhanced (predicted) ability to produce selected SCFAs, albeit *via* different metabolic pathways (Fig 4C). Together, these findings point towards a likely functional link between *Schistosoma* infection and the production of selected microbial metabolites as result of the dynamic cross-kingdom interaction in the gut.

## Enhanced potential for L-tryptophan biosynthesis and butyrate production in both schistosome-infected WT and HMA mice

*S. mansoni* infection was shown to significantly impact gut microbial amino acid metabolism in both WT and HMA hosts (S5 and S6 Tables), although several pathways changed in

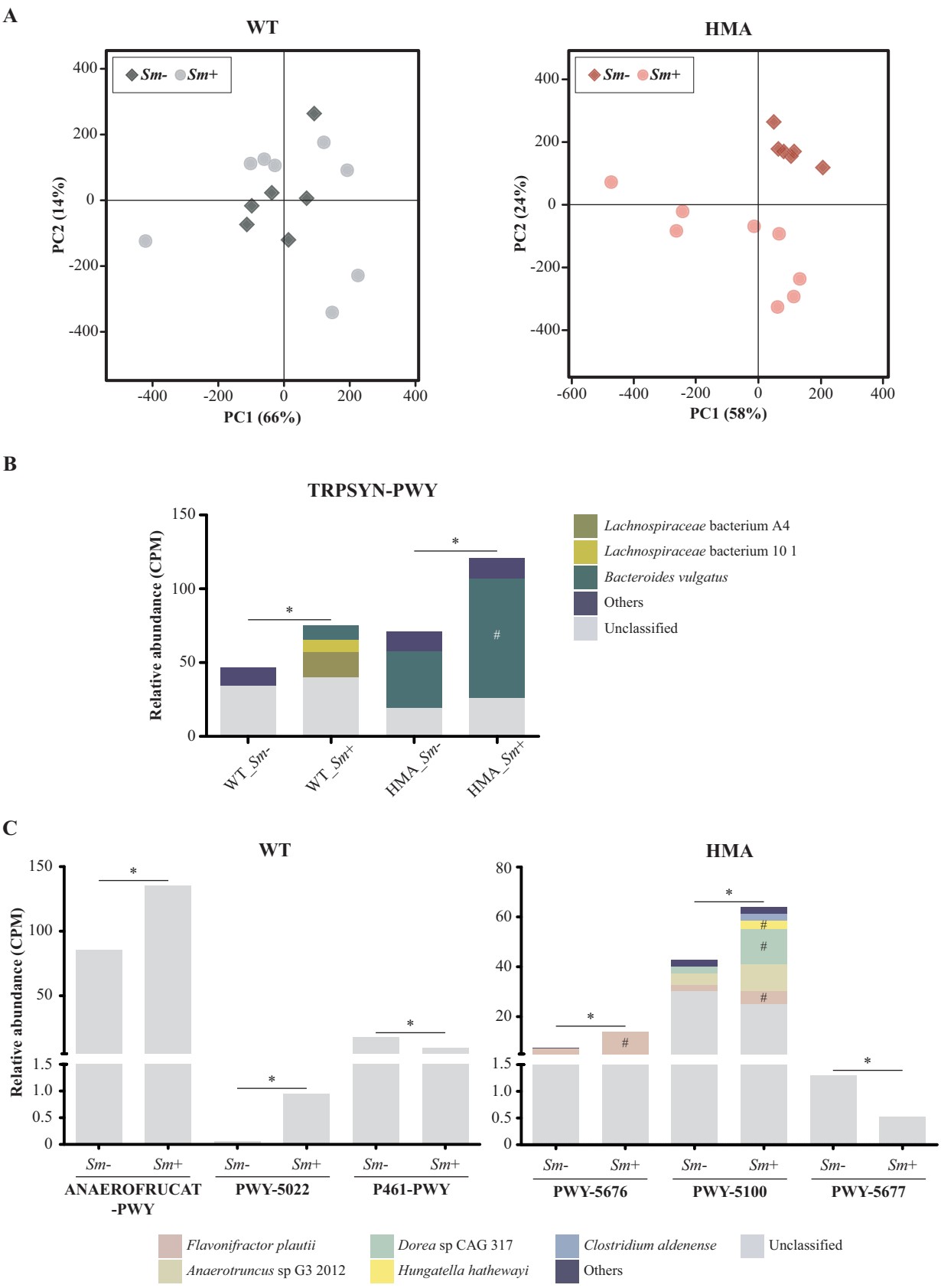

**Fig 4. Impact of *Schistosoma mansoni* infection on the fecal metabolic profiles of wild type (WT) and human-microbiota-associated (HMA) mice.** Principal Component Analysis (PCA) according to infection status (A) and infection-associated changes in gut metagenome capacity for tryptophan biosynthesis (B) and short-chain fatty acids production (C). Asterisks indicate significant differences in pathway abundance between *S. mansoni*-infected (*Sm+*) and uninfected controls (*Sm-*) of each mouse line by both MaAsLin2 ($p < 0.05$) and LEfSe [LDA score (log) $> 2$]. Hashtags indicate significant differences in the contribution of selected bacterial species to the overrepresentation of corresponding pathway(s) (MaAsLin2, $p < 0.05$). Full pathway denominations: TRPSYN-PWY = L-tryptophan biosynthesis; ANAEROFRUCAT-PWY = homolactic fermentation; PWY-5022 = 4-aminobutanoate degradation V; P461-PWY = hexitol fermentation to lactate, formate, ethanol and acetate; PWY-5676 = acetyl-CoA fermentation to butanoate II; PWY-5100 = pyruvate fermentation to acetate and lactate II; PWY-5677 = succinate fermentation to butanoate.

opposite directions following parasite colonization in each mouse line (S7 Fig). In particular, 'superpathway of L-serine and glycine biosynthesis I' (SER-GLYSYN-PWY), 'L-ornithine biosynthesis II' (ARGININE-SYN4-PWY), and 'GABA shunt' (GLUDEG-I-PWY; a pathway for L-glutamate degradation) were enriched in the microbiome of WT *Sm+* compared with WT *Sm-*, and in HMA *Sm-* compared with HMA *Sm+*. In addition, 'L-methionine biosynthesis III' (HSERMETANA-PWY) and 'L-arginine biosynthesis II (acetyl cycle)' (ARGSYNBSUB-PWY) were significantly enriched in HMA *Sm+* (S7 Fig). In contrast, 'L-tryptophan biosynthesis' (TRPSYN-PWY) was consistently linked to *S. mansoni* infection in both WT and HMA mice (Fig 4B). Expansion of this pathway in WT *Sm+* mice was linked to two *Lachnospiraceae* bacteria (i.e., A4 and 10 1) as well as to *Bacteroides vulgatus*; nonetheless, the individual contribution of these bacteria to the overrepresentation of the pathway was not statistically significant (Fig 4B). In HMA, in contrast, expansion of TRPSYN-PWY following infection was significantly linked to *B. vulgatus*, together with other low abundant bacteria (Fig 4B).

The microbial metabolism of amino acids plays key roles in the maintenance of gut barrier integrity and immune homeostasis [40]. For instance, polyamines derived from arginine and ornithine metabolism enhance gut barrier function by inducing overexpression of tight junction proteins and reducing inflammation [41–43]. For these reasons, an enhanced metabolism of these amino acids by the gut microbiome has been linked to a 'healthy' gut [44,45]. Given the differences detected between WT *vs*. HMA hosts in infection-associated alterations in arginine/ornithine/polyamine metabolism (S7 Fig and S5 and S6 Tables), caution is warranted when discussing any role(s) of these amino acids and corresponding metabolites in the context of schistosomiasis. However, the enhanced capacity for bacterial tryptophan biosynthesis detected in both rodent lines following *S. mansoni* infection may suggest a likely link between parasite infection and luminal production of this essential amino acid that, under physiological conditions, is mainly of dietary acquisition [46]. Both host- and microbiota-derived tryptophan metabolites are involved in the regulation of several host functions, including immune regulation by RORγt(+) IL-22(+) type 3 innate lymphoid cells (ILC3s) and maintenance of gut barrier function [46,47]. Hence, the enrichment of the 'L-tryptophan biosynthesis' pathway in the gut microbiomes of both WT and HMA mice following *S. mansoni* infection raises the question of whether this might serve as a compensatory mechanism to counteract local tissue damage caused by the migration of schistosome eggs through the intestinal wall. Future studies, including targeted perturbation of the 'L-tryptophan biosynthesis' pathway [e.g., 48] in the context of schistosome infection are needed to address this question.

Only a few gut resident bacteria, including selected species of the phyla Firmicutes and Bacteroidetes, are currently known to metabolize luminal tryptophan into indole and its derivatives [49]; these molecules may promote intestinal epithelial cell (IEC) renewal, barrier integrity, and maintenance of immune homeostasis *via* the activation of aryl hydrocarbon receptor (AhR) signaling in intestinal epithelial and mucosal immune cells [46,47]. Our analysis did not detect any pathway for tryptophan degradation; nevertheless, significant expansions of the tryptophan metabolizing species *P. distasonis*, *L. reuteri* and *L. johnsonii* [49] were

observed in the gut of WT mice following infection by *S. mansoni* (Fig 3C). Interestingly, populations of *L. reuteri*, but not of *L. johnsonii*, are known to expand in response to increased luminal tryptophan availability, and catabolize tryptophan into indole derivatives *via* aromatic amino acid aminotransferase [50]. In turn, *L. reuteri* mediates reprograming of intraepithelial CD4+ T cells into immunoregulatory CD4+CD8αα+ T cells *via* production of indole derivatives and activation of AhR signaling [51], and induces innate IL-22 production in the mouse gut by ILC3s, thus promoting gut barrier function [50]. IL-22 regulates production of antimicrobial peptides (AMPs) in the gut [52]. Strikingly, antibody-mediated neutralization of this cytokine was associated with the expansion of gut populations of *B. acidifaciens* in a murine model of *Salmonella* infection, an effect that was linked to an altered production of IL-22-regulated AMPs (i.e., Reg3β) [20]. In a different study, however, IL-22-mediated AMP responses led to a significant reduction of *Bacteroides* and the expansion of *Lactobacillus* [53]. Our finding of a concomitant expansion of *B. acidifaciens* along with several tryptophan metabolizing bacteria (i.e., *P. distasonis*, *L. reuteri* and *L. johnsonii*) in the gut of WT *Sm+* reveals a complex scenario, where both parasite-specific- and microbiota-modulated-immunity are likely to influence populations of selected microbial species. Future studies addressing the contribution of selected members of the gut microbiome to the local immune response during intestinal schistosomiasis will assist to disentangle the intricate interplay between parasite infection, these bacteria, and host immunity.

In addition to tryptophan metabolites, microbiota-derived SCFAs, and butyrate in particular, are involved in the regulation of AhR and its target genes in both liver and intestine [54,55]. In addition, butyrate promotes gut barrier integrity by modulating transcription factors in IECs (i.e., activating HIF1, STAT3 and SP1, and inhibiting NF-kB) and mediates anti-inflammatory functions by signaling on innate and adaptive immune cells [19]. Metabolic pathways involved in the production of butyrate were overrepresented in the gut microbiome of both WT and HMA mice following infection with *S. mansoni*. In particular, the pathways 'acetyl-CoA fermentation to butanoate II' (PWY-5676) and '4-aminobutanoate degradation V' (PWY-5022) were enriched in the gut microbiomes of HMA *Sm+* and WT *Sm+* mice, respectively, compared to their uninfected counterparts (Fig 4C and S5 and S6 Tables). The gut microbiome of WT *Sm+* was also enriched in the 'GABA shunt' pathway, whose products (i.e., 4-aminobutanoate and succinate) are metabolized to butyrate through the '4-aminobutanoate degradation V' pathway (cf. S5 Table). Furthermore, the gut microbiomes of infected animals of both host lines displayed enhanced potential for production of acetate and/or lactate, that are both key substrates for butyrate-producing bacteria [13,14,56,57]. In particular, the pathways 'pyruvate fermentation to acetate and lactate II' (PWY-5100) and 'acetylene degradation (anaerobic)' (P161-PWY) were enriched in HMA *Sm+*, whereas 'homolactic fermentation' (ANAEROFRUCAT-PWY) was overrepresented in WT *Sm+* (Fig 4C and S5 and S6 Tables). Specific bacteria linked to these functions remained largely unclassified both in WT and HMA mice (Fig 4C). Nevertheless, a significant contribution of *Flavonifractor plautii* to PWY-5676 was observed in HMA *Sm+*, whereas *Dorea* sp CAG 317, *F. plautii* and *Hungatella hathewayi* were significantly linked to the overrepresentation of PWY-5100 in the gut of the same animals (Fig 4C).

Additional pathways related to SCFA production were underrepresented in both WT *Sm+* and HMA *Sm+*, compared to uninfected counterparts. In particular, the pathways 'hexitol fermentation to lactate, formate, ethanol and acetate' (P461-PWY) and 'succinate fermentation to butanoate' (PWY-5677) were enriched in the gut microbiomes of WT *Sm-* and HMA *Sm-*, respectively (S5 and S6 Tables). Nonetheless, these pathways were comparatively less abundant than those involved in the same metabolic processes (i.e., lactate/acetate and butyrate biosynthesis) and that were overrepresented in *Sm+* of the corresponding host line (Fig 4C).

Metabolic profiling of biofluids had already revealed alterations of gut microbial metabolites in mice infected with *S. mansoni*, including SCFAs [58,59]. In particular, acetate, propionate and butyrate were depleted in urine samples of infected mice at 7 weeks post-infection, which suggested either a reduced production of these SCFAs by gut bacteria or an increased utilization of these metabolites by the vertebrate host [58]. In support of the latter hypothesis, in a separate study, significantly increased levels of propionate were detected in feces of *Sm+* between 7 and 9 weeks post-infection, although no statistically significant differences in butyrate levels were detected between infected mice and uninfected controls [59]. The discrepancies between data from our study and that by Li et al. [59] might be linked to differences between mouse strains [C57BL/6 *vs*. NMRI, respectively] and/or other factors known to affect the composition of the host gut microbiota, such as age or diet [18]. Nonetheless, observations from both our study and that by Li et al. [59]. support a link between *S. mansoni* infection and enhanced SCFAs biosynthesis in the rodent gut.

## Schistosomiasis may be associated with reduced horizontal gene transfer within the microbiota of HMA mice

The relative abundances of the top 10,000 functionally annotated gene families identified by HUMAnN3 in at least 5 metagenomes of either HMA or WT were compared between *Sm+* and *Sm-* by MaAsLin2 and LEfSe. The substantial differences in the microbiomes of WT mice between the two independent experiments performed ('B1' and 'B2'; cf. Materials and Methods) impaired the optimal fitting of the linear model (LM) calculated by MaAsLin2 to determine associations between gene family abundances and *S. mansoni* infection in these mice. Therefore, in order to overcome this technical limitation, the MaAsLin2 output was compared with that obtained by LEfSe; this approach led to the identification of 3 gene families whose relative abundances were altered by *S. mansoni* infection according to both MaAsLin2 (q<0.05) and LEfSe [LDA score (log)>2]. These gene families, enriched in the gut microbiome of WT *Sm+* mice compared to WT *Sm-*, encoded for 2 proteins involved in chromosome segregation (i.e., chromosome partitioning and ParB-like partition proteins) and a TraG-like protein involved in conjugative plasmid transfer (S8 Table). A total of 6 and 62 gut microbial gene families were enriched in HMA *Sm+* and HMA *Sm-*, respectively, by both MaAsLin2 (q<0.01) and LEfSe [LDA score (log)>2)] (S9 Table). Notably, amongst the latter, there were several families with roles in plasmid biology, including genes encoding proteins involved in conjugative plasmid transfer (e.g., MobA, MobB and MobC, amongst others), plasmid recombination (i.e., plasmid recombination enzyme family proteins), initiation of plasmid replication and plasmid copy control (e.g., protein involved in initiation of plasmid replication and RepA) (S9 Table).

Plasmids mediate horizontal gene transfer (HGT) between bacteria [60]. A recent study has revealed that HGT occurs frequently within individual human gut microbiomes, although frequency is higher in industrialized urban populations than in non-industrialized rural communities [61]. The specific factors responsible for such differences are currently unknown; however, it has been speculated that alterations in the gut ecosystem linked, for instance, to the dramatic changes in dietary and sanitation habits experienced throughout the industrialization process, rather than variations in bacterial species composition, are likely to be held accountable [61]. Improved sanitation in tropical and subtropical impoverished areas of the world has led to a significantly reduced risk of infection with many parasitic worms, including *Schistosoma* spp. [62]. Hence, the significance of our observation that several gene families related to regulation of plasmid transfer were underrepresented in the gut of HMA *Sm+* mice is worth exploring. It is well established that, within the gut microbiota, antimicrobial resistance

(AMR) genes are maintained and exchanged *via* HGT among bacterial communities [63]. A recent report showed no significant association between gut microbiota, AMR genes and uro-genital schistosomiasis in preschool-aged children [64]. However, further studies are needed to determine whether hepato-intestinal schistosomiasis might represent an additional factor limiting HGT, thus impacting on the distribution of AMR genes in the host gut. Interestingly, we identified a number of gene families related to resistance against several classes of antibiotics, including beta-lactams, fluoroquinolones, tetracyclines, aminoglycosides, macrolides and chloramphenicol (S10 Table) and detected significant differences (by MaAsLin2 only) in the abundance of 9 families linked to AMR in HMA mice (S9 Table). In particular, these gene families, all of which were significantly associated with the gut microbiome of HMA *Sm-* mice, encoded for proteins involved in resistance against beta-lactams (4 families), macrolides (3 families), tetracyclines and chloramphenicol (1 family each).

## Concluding remarks

In this study, we investigated the impact of *S. mansoni* infection on the gut microbial composition and predicted functions of two mouse lines with significant differences in their baseline gut microbial communities (i.e., WT and HMA). Comparisons of gut microbial functional profiles between *Sm+* and *Sm-* mice revealed infection-associated alterations in both hosts, some of which were consistently associated with worm colonization, whereas others displayed opposite trends in each infection model. Occurrence of consistent microbiome alterations in response to infection suggest a fine crosstalk between *S. mansoni* and the host gut microbiome. In particular, our findings support the hypothesis that gut microbial responses to *S. mansoni* infection might involve enhanced production of tryptophan metabolites and butyrate, and subsequent activation of AhR signaling and further butyrate-regulated pathways, in order to limit helminth-induced tissue damage. These findings offer a novel and intriguing perspective on the role(s) of the host gut microbiome in the pathophysiology of schistosomiasis, including its likely contribution to *Schistosoma* egg-induced intestinal pathology and inflammation in infected hosts [4,6]. Indeed, together with those previous reports, our current results point toward a likely dual role of the gut microbiome in the pathophysiology of schistosomiasis, where intestinal bacteria may contribute to egg-associated intestinal pathology while, simultaneously, protect the host from excessive tissue damage. This data may open new avenues towards the discovery of tentative bacterial targets for innovative control strategies, that are desperately needed for this neglected tropical disease. In addition, our preliminary observation that schistosomiasis mansoni might impact HGT in the vertebrate gut microbiota offers an intriguing insight into helminth-microbiota relationships, and calls for future investigations aimed to assess the role(s) that helminth infections might play in gene flow and spread of AMR in both rodent models and natural hosts.

## Materials and methods

### Ethics statement

The life cycle of *S. mansoni* (NMRI strain) was maintained at the Wellcome Sanger Institute (WSI) by breeding and infecting susceptible intermediate (*Biomphalaria glabrata* snails) and definitive hosts (female TO mice). All experimental infections and regulated procedures described in this study were approved by the Animal Welfare and Ethical Review Body (AWERB) of WSI. All experiments were conducted under Home Office Project Licenses (Procedure Project License—PPL) No. P77E8A062 held by Gabriel Rinaldi, and No. P6D3B94CC held by Trevor D. Lawley. The AWERB is constituted as required by the UK Animals (Scientific Procedures) Act 1986 Amendment Regulations 2012.

## Experimental procedures

Detailed methods for generating and breeding HMA mice, production of *S. mansoni* infective stages, experimental infections and parasitological procedures are available from Cortés et al. [8]. Briefly, 200 μl homogenates obtained from fresh feces from a healthy human donor were administered to 5 male and 5 female C57BL/6 germ-free mice by oral gavage once a week, for 3 weeks. Mice were subsequently set up as breeding pairs in a decontaminated positive pressure isolator. HMA mice used in this study belonged to the fourth generation of breeding animals; these were removed from the isolator in sealed ISOcages and maintained on a positive pressure ISOrack (Tecniplast) for use in the experiments described below. A total of 8 HMA and 8 WT mice (female C57BL/6, bred at the WSI) were infected percutaneously with 80 *S. mansoni* cercariae (*Sm+*), as described previously [65] and maintained for 50 days with access to food and water *ad libitum*. Six HMA and WT mice, respectively, were kept uninfected and used as negative controls (*Sm-*). Two independent experiments were performed (identified as 'B1' and 'B2', respectively) using identical procedures as described above. Experiment B1 included 5 *Sm+* and 3 *Sm-* of each HMA and WT, while experiment B2 included 3 *Sm+* and 3 *Sm-* of each HMA and WT [8]. All mice were euthanized 50 days post-infection, and adult worms recovered from *Sm+* mice *via* portal perfusion [65]. Control *Sm-* mice were perfused as the infected animals. Fecal pellets were collected directly from the colons of individual mice at necropsy; pellets were transferred to sterile tubes, snap frozen on dry ice, and stored at -80˚C until DNA isolation, which was performed within 3 weeks from sample collection.

## DNA extraction and shotgun metagenomics sequencing

Genomic DNA was isolated from a total of 28 colonic content samples and 4 no-DNA template negative controls from both independent experiments, using the PowerSoil DNA Isolation Kit (QIAGEN) according to manufacturers' instructions. DNA was prepared and sequenced at WSI using the Illumina Hi-Seq platform with library fragment sizes of 200–300 bp and a read length of 100 or 125 bp, as previously described [66].

## Bioinformatics analyses, statistics and reproducibility

WGS reads were subjected to quality and adapter trimming using Trim Galore (version 0.4.0) using default parameters, and paired reads of <20 bp for at least one of the two sequences were removed (https://github.com/FelixKrueger/TrimGalore; [67]). Quality-filtered reads were mapped against the *Mus musculus* reference genome (GRCm38 assembly) using the BWA-MEM algorithm [68].

Clean reads were mapped against reference bacterial genomes available from GenBank (S11 Table) and, for each sample, microbial community composition was defined based on the relative abundance of each genome displaying breadth coverage ≥1% and depth coverage >0.01X (S1 Table); the gut microbial communities of *Sm-* and *Sm+* HMA and WT mice were compared using the online software MicrobiomeAnalyst [69]. In particular, following CSS data normalization, samples were clustered by PCoA based on Bray-Curtis dissimilarities, and differences in beta diversity between groups were assessed by Analysis of Similarity (ANOSIM) [70]. Differences in microbial species richness and Shannon index between groups were assessed by unpaired t-test. Furthermore, heatmaps displaying Pearson correlation between community composition and explanatory variables, i.e., mouse line (WT and HMA), infection status (*Sm+* and *Sm-*) and experiment (B1 and B2) were generated; significant alterations in the relative abundances of microbial taxa following experimental *S. mansoni* infection were determined by LEfSe [71] and Mann-Whitney *U* test. Occurrence of linear relationships between the relative abundance (logarithmic) of gut bacterial species significantly impacted by *S. mansoni* infection, as well as infection burdens (i.e., worm counts and EPG of liver; cf. [8]),

was tested by Pearson correlation and the statistical significance of the Pearson correlation coefficient (*r*) was assessed by two-tailed t-test.

Functional annotations of gut metagenome data were performed using HUMAnN3 [72], ran from the biobakery/humann docker container (latest version as of October 2020) using the Chocophlan nucleotide database and Uniref90 protein database [73]. HUMAnN3 runs the Metaphlan program as an intermediate step to assign organism-specific functional profiling, which was performed using the developer-provided MetaPhlAn3 bowtie2 database [17,74]. Additional scripts embedded within HUMAnN3 (humann_rename_table and human_join_-table) were used to align gene family descriptions and merge the 128 original output gene family abundance tables into a single table. The HUMAnN3 pipeline also provided MetaCyc [75] pathway abundance detected per sample. HUMAnN's default Reads Per Kilobase (RPK) values for gene family and pathway abundances were transformed into copies per million (CPM) units using the "humann_renorm_table" script (included in the HUMAnN3 distribution; S10 and S12 Tables) and associations between the predicted functional properties of each metagenome and *S. mansoni* infection in either HMA and WT mice were assessed by PCA and CCA using Calypso [76]. Biological pathways and gene families significantly enriched in the metagenomes of either *Sm-* or *Sm+* mice (HMA and WT, respectively) were determined using MaAsLin2 [77], by applying the LM with default settings (except for the analysis of gene families, prior to which data were log-transformed); potential differences between samples collected in each independent experiment were taken into account by fixing this variable as a random effect. Furthermore, pathway and gene family relative abundances were compared between *Sm+* and *Sm-* of each HMA and WT using the LEfSe workflow [71] implemented in Galaxy (https://huttenhower.sph.harvard.edu/galaxy/), setting the variable "experiment" as subgroup for comparisons. Both pathway and gene family detection for a given rodent line was defined when confirmed in at least 5 out of the 14 samples analyzed per line.

## Supporting information

**S1 Fig. Whole genome sequencing output.** (A) Read counts obtained from fecal DNA extracts of wild type (WT) and human-microbiota-associated (HMA) mice, and negative controls (i.e., no-DNA template; NC); *p<0.05, **p<0.01 and ns = not significant by *post hoc* Dunn's test. (B) and (C) Base counts prior to (= sequenced) and following quality filtering and removal of host sequences (= filtered) according to mouse line and infection status, respectively. Differences between groups were assessed by Mann-Whitney *U* test. *Sm+* = *Schistosoma mansoni*-infected; *Sm-* = *S. mansoni*-uninfected.
(EPS)

**S2 Fig. Gut microbiota composition of wild type (WT) and human-microbiota-associated (HMA) mice.** (A) Venn diagram indicating the total number of bacterial species and corresponding genera identified in the metagenomes of WT and/or HMA mice following mapping of shotgun metagenomic reads against bacterial reference genomes available from the GenBank database. (B) Relative abundances of the 30 most abundant species.
(EPS)

**S3 Fig. Gut microbiota composition of wild type (WT) and human-microbiota-associated (HMA) mice.** Relative abundances, per sample, of the 30 most abundant bacterial species. Samples are hierarchically clustered according to Pearson correlation between species relative abundances, and explanatory variables are indicated at the top of the figure (i.e., mouse line, infection status and infection burdens; cf. Fig 1. EPG = eggs per gram).
(PDF)

**S4 Fig. Gut bacterial functional profiles in wild type (WT) *vs*. human-microbiota-associated (HMA) mice.** Venn diagram indicating the total number of bacterial metabolic pathways identified by HUMAnN3 in the gut metagenomes of at least five samples of WT and/or HMA mice.
(EPS)

**S5 Fig. Gut bacterial alpha diversity in *Schistosoma mansoni* infected (*Sm+*) and uninfected (*Sm-*) mice.** Shannon diversity and species richness in *Sm+* and *Sm-* samples of (A) wild type (WT) and (B) human-microbiota-associated (HMA) mice. Asterisks indicate significant differences between groups determined by unpaired t-test: *p<0.05.
(EPS)

**S6 Fig. Fecal metabolic profiles of wild type (WT) mice.** Principal Component Analysis (PCA) of the fecal metabolic profiles of WT mice according to experiment and infection status (*Sm+* = *Schistosoma mansoni*-infected; *Sm-* = uninfected controls).
(EPS)

**S7 Fig. Gut bacterial capacity for amino acid metabolism in wild type (WT) and human-microbiota-associated (HMA) mice.** For each mouse line, only differentially abundant pathways between *S. mansoni* infected (*Sm+*) and uninfected (*Sm-*) animals (cf. S5 and S6 Tables) are shown.
(EPS)

**S1 Table. List of bacterial genomes detected in fecal samples of wild type (WT) and/or human-microbiota-associated (HMA) mice by mapping of reads against reference bacterial genomes available from the GenBank database.** Only genomes displaying mean breadth coverage ≥1% and mean depth coverage >0.01X were considered in each mouse line.
nd = not detected.
(XLSX)

**S2 Table. Metabolic pathways.** (A) Metabolic pathways identified in at least 5 fecal samples of wild type (WT) and/or human-microbiota-associated (HMA) mice by HUMAnN3. (B) Metabolic pathways identified only in WT samples, stratified by known and unclassified organisms. (C) Metabolic pathways identified only in HMA samples, stratified by known and unclassified organisms.
(XLSX)

**S3 Table. Bacterial species in wild type (WT) and human-microbiota-associated (HMA) mice.** Results of LEfSe and Mann-Whitney *U* test applied to the identification of differentially abundant bacterial species between the gut microbial communities of *Schistosoma mansoni*-infected (*Sm+*) and uninfected (*Sm-*) WT and HMA mice.
(XLSX)

**S4 Table. Correlation between gut bacterial abundance and infection burdens.** Pearson correlation between the relative abundance of gut bacterial species significantly impacted by *Schistosoma mansoni* infection in wild type (WT) and human-microbiota-associated (HMA) mice (cf. Fig 3) and infection burdens (i.e., worm counts and eggs per gram (EPG) of liver; cf. [8]).
(XLSX)

**S5 Table. Metabolic pathways in wild type (WT) mice.** Differentially abundant metabolic pathways between the gut microbiomes of *Schistosoma mansoni*-infected (*Sm+*) and -uninfected (*Sm-*) WT mice by MaAsLin2 (p<0.05) and/or LEfSe [LDA score (log)>2]. For each pathway, mean abundance and standard deviation (SD), fold change between groups, and

functional classification according to MetaCyc database are provided. Fold change <0 = overrepresented in *Sm-*; fold change >0 = overrepresented in *Sm+*.
(XLSX)

**S6 Table. Metabolic pathways in human-microbiota-associated (HMA) mice.** Differentially abundant metabolic pathways between the gut microbiomes of *Schistosoma mansoni*-infected (*Sm+*) and -uninfected (*Sm-*) HMA mice by MaAsLin2 (p<0.05) and/or LefSe [LDA score (log)>2]. For each pathway, mean abundance and standard deviation (SD), fold change between groups, and functional classification according to MetaCyc database are provided. Fold change <0 = overrepresented in *Sm-*; fold change >0 = overrepresented in *Sm+*.
(XLSX)

**S7 Table. Metabolic pathways in both wild type (WT) and human-microbiota-associated (HMA) mice.** Differentially abundant metabolic pathways [MaAsLin2 p<0.05 and LefSe LDA score (log)>2] between the gut microbiomes of both WT and HMA mice infected by *Schistosoma mansoni* (*Sm+*) and matched uninfected controls (*Sm-*). For each pathway, functional classification according to MetaCyc database and fold change between infected and uninfected animals of each line are indicated. Fold change <0 = overrepresented in *Sm-*; fold change >0 = overrepresented in *Sm+*.
(XLSX)

**S8 Table. Gene families in wild type (WT) mice.** Differentially abundant gene families between the gut microbiomes of *Schistosoma mansoni*-infected (*Sm+*) and uninfected (*Sm-*) WT mice, by MaAsLin2 (q<0.05) and/or LefSe [LDA score (log)>2]. For each family, mean relative abundance, standard deviation (SD) and fold change between *Sm-* and *Sm+* are provided.
(XLSX)

**S9 Table. Gene families in human-microbiota-associated (HMA) mice.** Differentially abundant gene families between the gut microbiomes of *Schistosoma mansoni*-infected (*Sm+*) and -uninfected (*Sm-*) HMA mice, by MaAsLin2 (q<0.01) and/or LefSe [LDA score (log)>2]. For each family, mean relative abundance, standard deviation (SD) and fold change between *Sm-* and *Sm+* are provided.
(XLSX)

**S10 Table. Gene family abundances.** Relative abundances (in CPM; copies per million) of gene families computed by HUMAnN3 at community level for each sample included in this study.
(XLS)

**S11 Table. Mapping results.** Results of mapping of whole genome shotgun sequencing reads against reference bacterial genomes available from GenBank.
(XLSX)

**S12 Table. Metabolic pathway abundance.** Relative abundances [in CPM (copies per million) and for each sample included in this study] of metabolic pathways identified by HUMAnN3 at community level using gene abundances along with the structure of the pathway (based on MetaCyc pathway definitions).
(XLSX)

## Acknowledgments

The authors would like to thank Mandy Sanders (Wellcome Sanger Institute) for technical assistance.

## Author Contributions

**Conceptualization:** Alba Cortés, Gabriel Rinaldi, Cinzia Cantacessi.

**Data curation:** Alba Cortés, John Martin, Bruce A. Rosa.

**Formal analysis:** Alba Cortés, John Martin, Bruce A. Rosa, Klara A. Stark.

**Funding acquisition:** Alba Cortés, Matthew Berriman, Gabriel Rinaldi, Cinzia Cantacessi.

**Methodology:** Alba Cortés, Simon Clare, Catherine McCarthy, Katherine Harcourt, Cordelia Brandt, Charlotte Tolley, Gabriel Rinaldi.

**Project administration:** Gabriel Rinaldi, Cinzia Cantacessi.

**Validation:** Alba Cortés, John Martin, Bruce A. Rosa.

**Visualization:** Alba Cortés, John Martin, Bruce A. Rosa, Klara A. Stark.

**Writing – original draft:** Alba Cortés, Cinzia Cantacessi.

**Writing – review & editing:** Alba Cortés, John Martin, Bruce A. Rosa, Klara A. Stark, Simon Clare, Catherine McCarthy, Katherine Harcourt, Cordelia Brandt, Charlotte Tolley, Trevor D. Lawley, Makedonka Mitreva, Matthew Berriman, Gabriel Rinaldi, Cinzia Cantacessi.

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
