## [Decision Letter · Decision Letter 0]

27 Jul 2022

Dear Professor Cantacessi,

Thank you very much for submitting your manuscript "The gut microbial metabolic capacity of microbiome-humanized vs. wild type rodents reveals a likely dual role of intestinal bacteria in hepato-intestinal schistosomiasis" for consideration at PLOS Neglected Tropical Diseases. As with all papers reviewed by the journal, your manuscript was reviewed by members of the editorial board and by several independent reviewers. In light of the reviews (below this email), we would like to invite the resubmission of a significantly-revised version that takes into account the reviewers' comments. 

Please consider the suggested revisions for possible ways to improve your manuscript, including recommendations for additional tables/figures to improve clarity.

We cannot make any decision about publication until we have seen the revised manuscript and your response to the reviewers' comments. Your revised manuscript is also likely to be sent to reviewers for further evaluation.

Sincerely,

Jennifer A. Downs, M.D., Ph.D.

Academic Editor

Michael Hsieh

Section Editor

Please consider the suggested revisions for possible ways to improve your manuscript, including recommendations for additional tables/figures to improve clarity.

Reviewer's Responses to Questions

**Key Review Criteria Required for Acceptance?**

**Methods**

-Are the objectives of the study clearly articulated with a clear testable hypothesis stated?

-Is the study design appropriate to address the stated objectives?

-Is the population clearly described and appropriate for the hypothesis being tested?

-Is the sample size sufficient to ensure adequate power to address the hypothesis being tested?

-Were correct statistical analysis used to support conclusions?

-Are there concerns about ethical or regulatory requirements being met?

Reviewer #1: Th authors propose a hypothesis on the role of the intestinal microbiota in the pathogenesis of murine schistosomiasis in wild-type and microbiotme-humanized mice, and they test these hypotheses with adequate animal models, adequate infection models and adequate analyses. All methods are clrearly described in this or previous works (cited).

Ethical issues are addressed appropriately.

I cannot comment on the statistical analysis reg. sample size but the group sizes are in line with previous publications.

Reviewer #2: (No Response)

Reviewer #3: (No Response)

**Results**

-Does the analysis presented match the analysis plan?

-Are the results clearly and completely presented?

-Are the figures (Tables, Images) of sufficient quality for clarity?

Reviewer #1: The analyses match the proposed hypothesis and the proposed analytical models. The Results are well presented in both Figures and Tables (both of sufficient quality and presented very clearly), and a large body of suplementary data is provided for detailed information if required.

Reviewer #2: (No Response)

Reviewer #3: (No Response)

**Conclusions**

-Are the conclusions supported by the data presented?

-Are the limitations of analysis clearly described?

-Do the authors discuss how these data can be helpful to advance our understanding of the topic under study?

-Is public health relevance addressed?

Reviewer #1: The conclusions are fully supported by the data, and the authors fully acknowledge thelimitations of their analyes. They also indicate further hypotheses and research questions that should be addressed in future studies. The manuscript leads through the topic from the infection model to analytical details and clearly state that the focus of their conclusions is on the biochemical pathways that differend consistenly between infected and uninfected animals in both microbiome model, which complements previous studies focusing more on the bacterial communities themselves.

Reviewer #2: (No Response)

Reviewer #3: (No Response)

**Editorial and Data Presentation Modifications?**

Reviewer #1: The manuscript is clear and well presented, and I found only a single error; in the Introduction on page 7: The senctence starts with "Between 3 and 4 weeks" and "(S. mansoni and S. mansoni)" I believe should be changed to "(S. mansoni and S . japonicum)".

In the cited literature some articles (Frontiers series) are written in capital letters, this does not seem to be the citation style of the journal, but this will likely be changed during the final editing.

Reviewer #2: (No Response)

Reviewer #3: (No Response)

**Summary and General Comments**

Reviewer #1: This study describes changes in the microbiota upon Schistosoma mansoni infection in two different mouse models, a wild type and a humanized microbiome model. The results indicate that not only bacterial communities, but the microbial metabolome change with infection irrespective of the mouse strain (ruling out a possible influence of the host). The possible meanings of these findings are well discussed in relation to previous works on the topic of the rfole of the gut microbiome in infections with helminths, and help to form a more complete picture on the complex pathogenesis of chronic helminth infections and the interactions not only between host and parasite but involving the third party in this, the microbiome. The study is well designed, well presented and adds significant novel data to this complex topic.

Should shortening be required, the biology of Schistosoma (all three major species) included in the introduction probably does not need so much detail, however, the information is not completely redundant as pathogenesis is also the topic of the discussion.

Reviewer #2: This study from Cortes et al examines schistosome-induced changes to gut microbiome both in WT C57BL/6 mice and C57BL/6 reconstituted with a human microbiota. This builds on previous work from these groups showing Schistosoma mansoni infection alters microbial composition in Swiss-Webster mice (Jenkins Sci Rep 2018) and the two C57BL/6 models used in the current study (Cortes Front Immunol 2020, presumably using the same samples).The key difference this study uses whole genome sequencing rather than 16S rRNA analysis so giving species level resolution here. Whilst many of the headline findings have been reported in the previous studies (i.e. schistosome infection alters microbiota; this occurs in both WT and HMA mice; these two groups are very different), species level analysis now gives us a more in-depth picture of infection-induced changes and also allows a more extensive discussion of the implications (for instance, metabolic pathway analysis and potential effects on host immunity). Much of this is necessarily speculative but at the very least provides a set of testable hypotheses for future work.

Minor comments

- At times the analysis does feel a little overlong. For instance, the plasmid/horizontal gene transfer section is perhaps too speculative.

- Fig. 2C well shows clear separation between naïve and infected HMA mice (not the case for WT animals). It is not easy from Table S3/S4 to identify the pathways that drive this separation (whilst individual species are in Fig. 3). Similarly, a table showing the11 metabolic pathways that change in both WT and HMA (page 14) would be useful, highlighting the 3 that are consistent (and the 8 that are not). As it stands, TRPSYN-PWY is the standout discussed in depth.

- The 2020 Frontiers paper correlated changes in microbiota with parasite burden. Is it possible to do the same here? Assuming these are the same animals, there is a reasonable spread of worms (and eggs) within groups that may be informative, especially where hierarchical clustering (Fig. 1) groups some infected WT mice with naives – are these the ones with lower worm burdens? I expect this is too simplistic given the nature of the outliers, but would be good to see.

Reviewer #3: This study by Cortés et al presents an in-depth investigation of microbial traits associated with S. mansoni infection in two mice strains – a wild type strain and a strain with a humanized microbiota. They used shotgun sequencing to compare species-level taxonomic as well as functional profiles between controls and rodents infected with S. mansoni. The study provides new and original results and discussion points. However, the manuscript could potentially be improved by a few additional considerations described below:

Major:

C1: Decrease in gut bacterial richness in HMA at day 50 – I understand that it might be difficult (or impossible) but comparing the original donor sample to the engrafted community could also be of interest. Was the decrease in diversity also associated with a decreased in bacterial load?

C2: An additional figure (or supplementary figure) providing a visual comparison of the taxonomic composition for each individual mouse (e.g. barcharts) sorted according to their position in the heatmap cluster would complement the heatmap well. 

C3: Does the enrichment of individual SCFA production/metabolism-related pathways also reflect an enrichment at a higher metacyc functional level (e.g. Fermentation to Butanoate)? If yes, a comparison of the Sm-/Sm+ abundance ratios between mice strains could be interesting.

C4: The discussion about HGT/AMR is interesting. However, since the study was conducted using shotgun sequencing, more in-depth analyses could be performed using dedicated softwares to measure e.g. HGT rates (with metachip or daisysuite) or to investigate mechanisms of resistance more in detail (e.g. if plasmid driven or not?).

Minor:

C5: missing information about the sequencing results (e.g. sequencing depths etc..)

C6: “(S. mansoni and S. mansoni)”

C7: Section results/discussion could be separated for improved clarity.

C8: a visual representation accompanying the statements made about the differences in amino acid metabolism (end of page 14) would be useful.

C9: “Expansion of this pathway in WT Sm+ mice was linked to two Lachnospiraceae bacteria (i.e., A4 and 10 1) as well as to Bacteroides vulgatus; nonetheless, the individual contribution of these bacteria to the overrepresentation of the pathway was not statistically significant (Fig 4B). In HMA, in contrast, expansion of TRPSYN-PWY following infection was significantly linked to B. vulgatus, together with other low abundant bacteria (Fig 4B).”

Would it be possible to compare the actual genes B. vulgatus contributed to the pathways?

PLOS authors have the option to publish the peer review history of their article (what does this mean?). If published, this will include your full peer review and any attached files.

Reviewer #1: Yes: Anja Joachim

Reviewer #2: No

Reviewer #3: No
---

## [Decision Letter · Decision Letter 1]

9 Oct 2022

Dear Professor Cantacessi,

We are pleased to inform you that your manuscript 'The gut microbial metabolic capacity of microbiome-humanized vs. wild type rodents reveals a likely dual role of intestinal bacteria in hepato-intestinal schistosomiasis' has been provisionally accepted for publication in PLOS Neglected Tropical Diseases.

Best regards,

Jennifer A. Downs, M.D., Ph.D.

Academic Editor

Michael Hsieh

Section Editor

We appreciate the authors' excellent work on revisions and look forward to publishing this manuscript.

Reviewer's Responses to Questions

**Key Review Criteria Required for Acceptance?**

**Methods**

-Are the objectives of the study clearly articulated with a clear testable hypothesis stated?

-Is the study design appropriate to address the stated objectives?

-Is the population clearly described and appropriate for the hypothesis being tested?

-Is the sample size sufficient to ensure adequate power to address the hypothesis being tested?

-Were correct statistical analysis used to support conclusions?

-Are there concerns about ethical or regulatory requirements being met?

Reviewer #1: As stated in my previous evaluation, the objectives are clearly stated, the design and the applied methods are appropriate and ethics requirements are met.

Reviewer #2: (No Response)

Reviewer #3: The authors have carefully addressed my comments from the initial submission. I have no further comments.

**Results**

-Does the analysis presented match the analysis plan?

-Are the results clearly and completely presented?

-Are the figures (Tables, Images) of sufficient quality for clarity?

Reviewer #1: Ther results match with the methods and are clearly presented.

Reviewer #2: (No Response)

Reviewer #3: (No Response)

**Conclusions**

-Are the conclusions supported by the data presented?

-Are the limitations of analysis clearly described?

-Do the authors discuss how these data can be helpful to advance our understanding of the topic under study?

-Is public health relevance addressed?

Reviewer #1: The conclusions are supported by the results, and the authors are sufficiently critical towards their findings. All issues are addressed. The work contributes suginificantly to the understanding of the pathogeneesis of schistosomiasis.

Reviewer #2: (No Response)

Reviewer #3: (No Response)

**Editorial and Data Presentation Modifications?**

Reviewer #1: I have not found any points that need frther revision, and recommend to accept the manuscript in its current version.

Reviewer #2: (No Response)

Reviewer #3: (No Response)

**Summary and General Comments**

Reviewer #1: I already summarized my evaluation in the first review.

Reviewer #2: All comments well addressed, no further comments

Reviewer #3: (No Response)

PLOS authors have the option to publish the peer review history of their article (what does this mean?). If published, this will include your full peer review and any attached files.

Reviewer #1: **Yes: **Anja Joachim

Reviewer #2: No

Reviewer #3: No

---

## [Editor Report · Acceptance letter]

18 Oct 2022

Dear Professor Cantacessi,

We are delighted to inform you that your manuscript, "The gut microbial metabolic capacity of microbiome-humanized vs. wild type rodents reveals a likely dual role of intestinal bacteria in hepato-intestinal schistosomiasis," has been formally accepted for publication in PLOS Neglected Tropical Diseases.

Best regards,

Shaden Kamhawi

co-Editor-in-Chief

Paul Brindley

co-Editor-in-Chief
